# How Are Tier 2 Metropolises Affected by Housing Asset Value Deflation in the Depopulation Era? A Comparison between the Tokyo and Kansai Metropolitan Areas

Masaaki Uto [1,*] , Sophie Buhnik [2] and Yuki Okazawa [3]

1 Graduate School of Environmental Information Studies, Tokyo City University, Tokyo 1588557, Japan
2 Ecole Supérieure des Professions Immobilières, 92300 Paris, France; s.buhnik@groupe-espi.fr
3 Mitsubishi Research Institute Ltd., Tokyo 1008141, Japan
* Correspondence: mauto@tcu.ac.jp

**Abstract:** This study analyzes the differences and similarities between Tier 1 (Tokyo) and Tier 2 (Kansai) metropolitan areas due to shrinking city problems. Both metropolitan areas will see a dramatic decrease in the housing asset value (HAV). Kansai is declining at a faster pace than Tokyo: it is projected that HAVs will register a further decrease of around 38% by 2045, and the decline will be quantitatively more important in the northern suburbs of Osaka. These results raised the question of whether Kansai would be more impoverished by HAV deflation. By focusing on the income multiplier of HAV per household, we find that Tokyo has a higher income multiplier of around 4 (against 2 for Osaka), thus causing much greater HAV deflation per household in Tokyo. Greater HAV deflation per household entails more severe problems for elderly households that need to finance their retirement. Considering our findings, despite earlier and faster trends of HAV deflation in the Tier 2 metropolitan area, the Tier 1 metropolitan area could face big socioeconomic challenges in the future. We conclude that HAV deflation leads to problems of different nature depending on metropolitan rank, rather than just knowing which one is losing more through HAV deflation.

**Keywords:** asset values; depopulation; land prices; metropolises; shrinking cities; regression model

## 1. Introduction

Initially conceptualized to examine the decline of post-industrial Western cities, "shrinking cities" have become ubiquitous in the literature about Japan's urban dynamics in the 21st century [1]. Indeed, a majority of its localities has been coping with shrinkage for at least 15 years, leading to a multiplication of abandoned spaces and infrastructures [2–5]. Approximately three-quarters of all Japanese municipalities lost inhabitants between 2005 and 2020, and the few places that consistently gained residents are situated within the country's main metropolises. Even parts of Tokyo are shrinking, and there is "no truly booming city" in today's Japan [6]. The driver behind this phenomenon is depopulation: with an ongoing deficit of births relative to deaths not balanced with positive migration rates, the number of inhabitants may plunge from 127 million in 2008 to barely 100 million by 2050 according to the projections of the National Institute of Population and Social Security Research of Japan. COVID-19 has only accelerated the country's transition into degrowth [7]. Therefore, the central question of current research on Japanese cities is not to wonder whether they will shrink or not, but whether they will avoid full-scale collapse.

In this respect, Japan provides one of the most relevant contexts to examine "a future of globally-extensive decline" [8]. The correlation between depopulation, housing vacancy, and impoverishment has especially garnered the attention of geographers and economists. Their works highlight that a stronger pace of shrinkage systematically implies a further decrease in land prices [9–12]. An overall depreciation of the value of individually owned properties ensues. However, one limitation of existing the literature concerns its

geographical scope: the intensity of shrinkage and its impacts on the housing sector vary considerably from one locality to another [13–15], but Japanese regions have not received equal attention. Indeed, most studies that have categorized the factors leading to different trajectories of decline [16–18] or analyzed the impacts of shrinkage on the built environment and real estate markets [19–23] base their fieldwork upon cases situated either in Tokyo or a non-metropolitan municipality. Metropolitan-level studies centered on cities as big as Osaka and Nagoya are absent from this stream of investigations.

Our study fills this gap by clarifying municipal-level changes in housing asset values (HAV) in the Kansai metropolitan area (which overlaps the Osaka, Kyoto, Hyogo, Nara, and Wakayama Prefectures) (Figure 1). The Kansai metropolitan area has around 20 million habitants in 2020 and is ranked among the world's top 10 metropolitan areas, so we defined here such class of metropolitan area (like Dhaka, Beijing, and Mumbai) as Tier 2. Also, we defined Tier 1 as a city region with over 30 million habitants like the Tokyo metropolitan area, which had 36 million habitants in 2020, and thus remains the world's top rank metropolitan area (a threshold soon reached by Delhi and Shanghai) [24] (p.4).

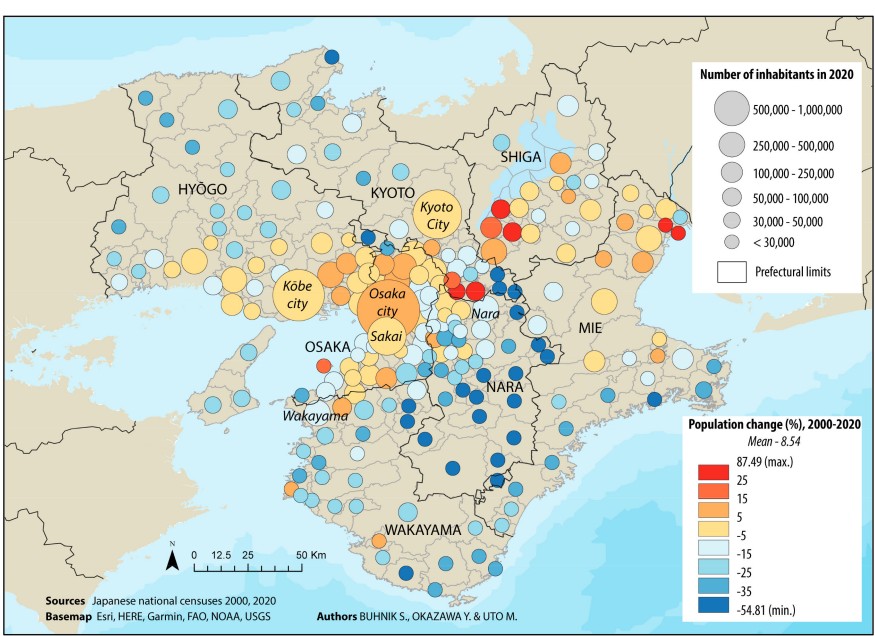

**Figure 1.** Population size by municipality in 2020 and rates of population change (2000–2020) in the prefectures covering the Kansai metropolitan area.

The Kansai metropolitan area accounts for a gradually decreasing percentage of Japan's population and GDP, which illustrates its difficulties to resist a polarization of political and economic power by Tokyo [25]. For these reasons, the inner–outer distribution of decline, obvious in Tokyo's case [26], is less evident here. These regional differences must be better understood if we want to improve the shrinking cities problem in Japan and beyond.

## 2. Literature Review

Japan's transition into aging and decline could be foreseen as early as in the 1980s [6]. But one could hardly predict that, out of a lack of demographic growth, the market downturn precipitated by the burst of the bubble in 1990 would be prolonged into a recession which the country unevenly recovered from [27,28]. According to data collected by the Ministry of Land, Infrastructure, Transport, and Tourism of Japan (MLIT), land prices of residential areas throughout the country in 2019 barely represented 50% of their 1989 nominal value. A state of the art about the geographies of territorial development and residential markets in Japan documents a transition from generalized growth to extensive decline, where three stages can be discerned.

Until the mid-1990s, skyrocketing land prices, combined with the spread of car-based mobilities, led to a peculiar "doughnut effect": households seeking access to homeownership had to go farther from downtown areas to purchase land, hence a demographic decline occurring inside and near the business districts of large cities [29]. The latter registered a rise in the day-to-night population ratios because employment, especially in the service sector and within the Tokyo region, did not relocate significantly to "edge cities" in spite of plans facilitating office dispersion [15,30]. Meanwhile, hollowing-out became visible in the cores of mid-size cities and capitals of non-metropolitan prefectures [14,20].

The early 2000s to mid-2010s witnessed a reversal in growth dynamics within metropolitan areas (Figure 1), one of the distinctive features of Japan's contemporary urban dynamics [8,11,31,32]. The game-changing Urban Renaissance Special Measures Law was introduced in 2002, with the aim of enhancing the global competitiveness of Japan's economy after a decade of stagnation [33,34]. It promoted transit-oriented redevelopment and large-scale renewal projects conducted within selected perimeters [35,36]. Thanks to a relaunch of condominium construction, the wards of cities with over 500,000 inhabitants began to attract migrants again, capitalizing on the newfound affordance of their housing supply for certain classes [25,37–39]. On the contrary, decline worsened in rural margins, regional cities (ranging between 50,000 and 400,000 inhabitants) and, within metropolitan regions, outlying municipalities, due to a "double demographic disequilibrium" [40]: out-migrations of working-age adults reinforce natural degrowth. The subsequent aging of households that remain in suburbs mirrors a restructuring of the intermingled political, economic, and social dimensions behind the post-war success of suburbia [8]. Frequently criticized because of their sprawl and bed town atmosphere, Japanese suburbs now come up as less adapted to a society where singles outnumber the male-breadwinner household [41–44].

The 2010s do not witness major shifts in the abovementioned trends. But the administrations' continuing "push towards fiscal devolution while carrying out focused urban revitalization" [45] contributed to the political construction of shrinkage as a spatial expression of Japan's widening inequalities [6]. First, the "back-to-the-city" movement, under the influence of real estate securitization, has bolstered Tokyo and the cores of a few metropolises [22,35]; property investments primarily target neighborhoods with high-end commercial and service activities. In mid-size cities, by contrast, the willful application of compact city schemes did not entail strong downtown revitalization, with few exceptions [18]. Second, discourses promoting a compact city as a solution to fight sprawl actually undermine the sustainable nature of many suburban or rural communities [46,47]. The 2014 Vacant Houses Special Measures Act has equipped localities with more tools to tear down derelict houses, but problems now lie in a lack of financial, technical, and human resources to achieve such targets [10]. Therefore, the fabric of Japan's cities displays intricate patterns of devitalization and dedensification [36,48]. As municipalities struggle to keep their number of taxpayers afloat, they welcome mixed-use projects that incorporate ecological and digital innovations, often several blocks away from obsolescent neighborhoods. Consequently, within any municipality, urban perforations born out of an accumulation of vacant lots coexist with scrap-and-build processes or new allotments [23]. There is thus a heightened risk of creating a regional oversupply of housing.

This supply/demand imbalance has soared in Japan's formerly booming suburbs [12,21,49] and causes the overall depreciation of property values [22]. Put in a comparative light, nonetheless, the correlation that the Japanese literature makes between depopulation, shrinkage, and land/real estate devaluation seems more straightforward than in other mature countries, where the "cold market" phenomenon exists but remains embedded in wider contexts of housing price inflation. In France for example, de-industrializing regions and rural margins that have undergone population decline and a rise in vacancy for decades can still register high levels of investment stimulated by a steady nationwide inflation of housing markets [50]. Meanwhile, the surge of foreclosures after 2008 in the USA hit the disadvantaged downtown areas and inner suburbs of post-Fordist cities. Abandonment of ownership rights nurtures a neglect of properties

and puts downward pressure on local housing values [51]. Yet, this depreciation precisely attracts acquirers who bargain on the profitability of "distressed real estate" [52]. Such opportunistic investments, encouraged by a scarcity of affordable housing on a national scale, have become rather rare in Japan. Despite the dearth of research about buyers of land and houses in depopulating areas, we can infer that their numbers are not enough to offset hollowing-out processes. As noted by [51], unmaintained empty houses generating negative externalities in Japan are not the outcome of foreclosures: it is the byproduct of inheritance issues. Children of owners are often not interested in moving into their parents' home after the latter have entered care facilities or passed away [53]. Barely 20% of heirs are willing to engage in rehabilitations of houses with almost no chance to be resold or rented [54].

To sum up, there is now a remarkable body of work about the intermingled issues faced by declining residential areas in today's Japan: their aging is a sign they are not among the most attractive places in a given city. They are subsequently at extreme risk of experiencing long-term vacancy and contribute to a decrease in the average value of existing assets within the city's boundaries. However, in spite of gradual attention paid to spatial variegations, most case studies, whether they adopt an inter-municipal approach or focus on one municipality for reiterated fieldwork, are overwhelmingly directed towards the biggest cities like Tokyo. So, we believe our study is remarkable in the shrinking cities study field.

## 3. Methods

### 3.1. Research Questions and Targeted Study Area

Ref. [55] predicted that the deflation of the HAV would reach 94 trillion JPY (667 billion US dollars) by 2045 for the Tokyo region and exceed 10 million JPY (around 71,000 USD, 1 USD = 141 JPY, as of December 2023) per household on average, with detrimental effects on citizens and urban planning in three ways: 1. the immobility of aging homeowners, who cannot rely on asset-based welfare logic to cover their care needs; 2. impediments to compact city policies that require resources to coordinate demolitions and relocations; 3. a drastic reduction in local government tax income. But what about metropolises like Osaka, Nagoya, or Fukuoka, whose size is on par with Paris or Chicago? Squeezed between a scholarly emphasis on Tokyo and a political prioritization of rural revitalization, they are quasi-absent from the literature about uneven spatial development in a post-growth context. Fleshing out their reconfigurations would yet provide original answers to these two questions:

(1) In Tier 2 metropolises (Kansai) compared to Tier 1 (Tokyo), are depopulation and decrease in HAV distributed in a similar pattern or not?

(2) In Tier 2 metropolises (Kansai), are the effects of HAV deflation more severe than Tier 1 (Tokyo), both in percentage and volume of financial loss?

Both Tokyo and Kansai qualify as large metropolitan areas with the country's highest levels of commuting. Tokyo had nearly 36 million inhabitants in 2022, against 19.2 for Kansai. There are 206 municipalities in Tokyo and 226 in Kansai. The cities of Kobe, Kyoto, and Nara, respectively, include 1.6 million, 1.47 million, and 367,000 inhabitants and are prefectural capitals, while Sakai (820,000) belongs to Osaka Prefecture (Figure 1).

Train commuting still amounts to around one half of daily trips in Kansai [56]. Figure 2 is a result of our measurements of average commuting times from each municipality's city halls to the stations closest to the CBDs (central business districts) of Osaka, Kyoto, and Kobe, relying on data from Google Maps (for car) and "Yahoo! Transfer Guide" (for train). We assumed that the majority of commuters use the main station of their municipality of residence (close to the city hall in general) at 9.00 a.m. on weekdays. Then, we added 10 min corresponding to average commuting time from home to the municipality's main station or the city hall (for travel by car), and 10 min for the travel from a CBD station to one's workplace.

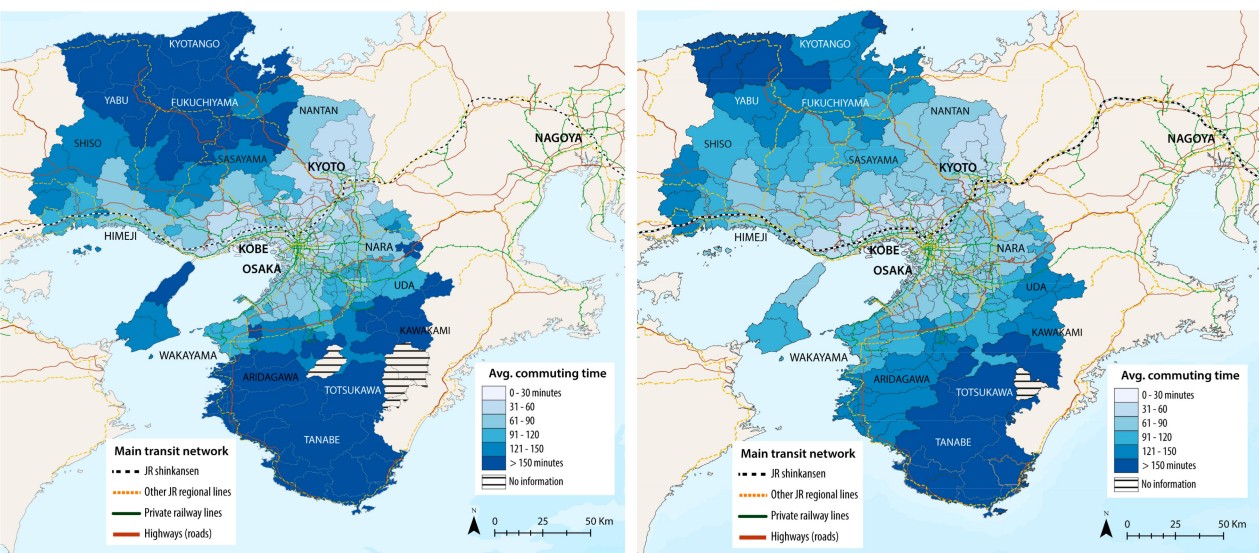

**Figure 2.** Average commuting time to the employment cores of Osaka from the municipalities of Kansai (in minutes), by train (**left**) and car (**right**), in 2023.

### 3.2. Approach Design

We applied a year-by-year municipal-level estimation of the impacts of population changes on HAVs from 2022 to 2045, which [55] formulated for a study of the Tokyo Metropolitan Area. First, we must project future land prices by municipality to examine how population decline affects housing land prices. Then, HAVs must be calculated on the basis of projected prices of land for housing. This is why we describe our method to project future housing land prices below.

The authors of [57] analyzed Japanese housing price determination mechanisms using an index of demographic factors. Their method is composed of two steps. In the first step, they estimate the age-specific demand for housing using micro data from the National Survey of Family Income and Expenditure and constructed an aggregate housing demand variable. In the second step, the housing stock and housing price equations are estimated on the basis of the relationship between the demand of each household and the housing market as a whole. But such microdata is not open to the public, so we cannot obtain the same datasets. Authors of [58,59] estimated land prices in year *t* in region *I* determined by Equation (1):

$$\ln P_{it} = \alpha_i + \beta_1 \ln GDPPC_{it} + \beta_2 \ln OLDDEP_{it} + \beta_3 \ln TPOP_{it} + e_{it} \tag{1}$$

$P_{it}$ : Publicly assessed land prices (residential) in region *i* in year *t*;
$GDPPC_{it}$ : Per capita GDP in region *i* in year *t*;
$OLDDP_{it}$ : Shares of 65-and-older population and 15–64 population in region *i* in year *t*;
$TPOP_{it}$ : Total population in region *i* in year *t*.

The authors then projected the impact of future changes in population distribution on changes in real estate prices in each region using the estimation formula as in Equation (2):

$$\Delta \ln P_{it} = a_i + b_1 \Delta \ln GDPPC_{it} + b_2 \Delta \ln OLDDEP_{it} + b_3 \Delta \ln TPOP_{it} + ECT_{it-1} + v_{it} \tag{2}$$

$ECT_{it-1}$ is the error correction term, defined by

$$ECT_{it} \equiv \ln P_{it} - (\alpha_i + \beta_1 \Delta \ln GDPPC_{it} + \beta_2 \Delta \ln OLDDEP_{it} + \beta_3 \Delta \ln TPOP_{it})$$

However, Ref. [60] verified the hypothesis of [61], which states that due to the durability of housing, the stock remains elastic when the population is increasing and inelastic when the population is decreasing. That is, using the data about individual prefectures,

they conducted estimates with separate parameters both for when the population is increasing and when it is decreasing. This paper estimates that during the period 1970–2000, the parameter for when the population is increasing is positive and statistically significant and OLDDP is negative and statistically significant; however, during the period 2000–2015, only the parameter for when the population is decreasing is positive and statistically significant. The authors also determined that elasticity was higher for declining rather than increasing population changes.

Here, we attempted to revalidate Equation (2) adopted from [58,59] based on the implications of [61]. However, we do not adopt GDPCC as we cannot use continuous GDP data by municipality. Specifically, in the second term of Equation (2), the same exponent applies to the growth of both populations: 15–64 and 65 and older, thereby making it impossible to reflect differences in the impacts on prices when the population fluctuates by age group. As a result, we attempted to modify the individual exponents of the two populations in the second term, as in Equation (3):

$$
\begin{aligned}
\Delta \ln P_{it} &= a_i + ln\frac{\left(POP_{it}^{65}/POP_{it-1}^{65}\right)^{b_2}}{\left(POP_{it}^{15-64}/POP_{it-1}^{15-64}\right)^{\beta_2}} + b_3 ln\frac{TPOP_{it}}{TPOP_{it-1}} + ECT_{it-1} + v_{it} \\
&= a_i + ln\frac{\left(POP_{it}^{65}/POP_{it-1}^{65}\right)^{b_2}}{\left(POP_{it}^{15-64}/POP_{it-1}^{15-64}\right)^{b_2}}\frac{1}{\left(POP_{it}^{15-64}/POP_{it-1}^{15-64}\right)^{\beta_2-b_2}} + b_3 ln\frac{TPOP_{it}}{TPOP_{it-1}} + ECT_{it-1} + v_{it} \\
&= a_i + b_2 ln\frac{POP_{it}^{65}/POP_{i5-1}^{65}}{POP_{it}^{15-64}/POP_{it-1}^{15-64}} - (\beta_2 - b_2)ln\frac{POP_{it}^{15-64}}{POP_{it-1}^{15-64}} + b_3 ln\frac{TPOP_{it}}{TPOP_{it-1}} + ECT_{it-1} + v_{it}
\end{aligned}
\tag{3}
$$

$POP_{it}^{65}$ : 65-and-older population in region *i* in year *t*;

$POP_{it}^{15-64}$: 15–64 population in region *i* in year *t*;

Here, as the working-age population is expected to contribute to increasing housing demand, it might be assumed that $-\beta_2 + b_2 > 0$. The following relationship has been established in Japan to date: $\frac{POP_{it}^{15-64}}{POP_{it-1}^{15-64}} \cong \frac{TPOP_{it}}{TPOP_{it-1}}$

For example, while the total population from 2000 to 2005 increased by 0.7%, the population aged 15–64 years decreased by 2.4%, and those aged 65 and older increased by 16%. From 2005 to 2010, the same growth rates were 0.2% increase, 0.4% decrease, and 14% increase, respectively, and from 2010 to 2015, they were 0.8% decrease, 6% decrease, and 14% increase. Accounting for this, it is possible to restate the formula as in Equation (4):

$$
\Delta ln P_{it} = a_i + b_2 ln\frac{POP_{it}^{65}/POP_{i5-1}^{65}}{POP_{it}^{15-64}/POP_{it-1}^{15-64}} + (-\beta_2 + b_2 + b_3)ln\frac{TPOP_{it}}{TPOP_{it-1}} + ECT_{it-1} + v_{it}
\tag{4}
$$

In this case, the parameter in the third term increases, and much of the price change can be explained by changes in the total population. According to [60], loss in the statistical significance of the second term parameter from the 2000s may be due to the population aged 15–64 years entering a phase of decline. As a result, the possibility of being able to explain much of the change in land prices by changes in the total population will likely strengthen.

In other words, if the working-age population has entered a phase of decline and changes in the working-age population and total population are similar, land prices will be more impacted by changes in the total population than by changes in the age distribution of that population. These changes help forecast much of the change in land prices using changes in the total population of Japan.

*3.3. Methods*

Our discussions up to this point have assumed that land prices respond more sensitively to phases of population decline than to phases of population increase. We attempt to

verify this here. We estimated the following land price function based on publicly assessed land prices for the municipalities of Kansai and population data as shown in Equation (5):

$$\frac{P_{it}-P_{it-1}}{P_{it-1}} = b_1 * DUM^+ * \frac{TPOP_{it}-TPOP_{it-1}}{TPOP_{it-1}} + b_2 * DUM^- * \frac{TPOP_{it}-TPOP_{it-1}}{TPOP_{it-1}} + b_3 * DUM^{big} * lnPrice_{i2000} + b_4 * DUM^{small} * lnPrice_{i2000} + v_{it} \tag{5}$$

$DUM^+$: Dummy variable takes a value of 1 when the total population change is zero or positive and 0 in other cases;
$DUM^-$: Dummy variable takes a value of 1 when the total population change is negative and 0 in other cases;
$DUM^{big}$: Dummy variable takes a value of 1 when the region *i* population is over 500 thousand and 0 in other cases;
$DUM^{small}$: Dummy variable takes a value of 1 when the region *i* population is under 500 thousand and 0 in other cases;
$Price_{i2000}$: Land price in 2000 in region *i*.

Only population data have been officially published by the Japanese government in the form of long-term projections running until 2045, for all municipalities. These demographic projections are relatively accurate because only natural and social factors affect future population changes. Population projections made by the National Institute of Population and Social Security Research of Japan, and endorsed by the Japanese government, are considered accurate because they are based on assumptions about trends in fertility, mortality, and international migration that are regularly revised. That is why we devised a simple estimation model that used only the officially projected increase and decrease in population.

We did not add the error correction term ($ECT_{it-1}$ in equation [4]), because we estimate the long-run effects of population change. For example, [57] distinguished the long-run and short-run effects. They found a positive effect of housing stock on price by error correction model. And they concluded that effects can be thought of as the result of the short-run inelastic supply of housing. Therefore, we added $Price_{i2000}$ to control the endogenous social and economic conditions by municipality, because we cannot use the continuous GDP data. In Japan, a municipality with a population over 500 thousand usually becomes a "City designated by government order" or a "Special ward" that is distinguished from other municipalities. We used $DUM^{big}$ and $DUM^{small}$ to control the municipality position.

$DUM^+ * \frac{TPOP_{it}-TPOP_{it-1}}{TPOP_{it-1}}$ and $DUM^- * \frac{TPOP_{it}-TPOP_{it-1}}{TPOP_{it-1}}$ represent the rate of change in land price when the total population increases and decreases, respectively. According to [61], the sensibility of the rate of change in land price differed depending on whether the population increased or decreased, so that both of them are variables independent from each other. In addition, $DUM^{big} * lnPrice_{i2000}$ and $DUM^{small} * lnPrice_{i2000}$ are also independent variables representing the rate of change in land price for different municipality positions, as [57] distinguished them. Hence, the multicollinearity does not occur in Equation (5).

We estimated the explained and explanatory variables, land price changes, and population changes using the data not only for a given year but also for the five- to seven-year moving averages as in Table 1, in order to observe short- and medium-term impacts of population changes on land prices. The estimation period is from 2000 to 2021. According to [60], land prices were impacted more by changes in the total population than by changes in the age distribution of that population in the 2000s.

The results are shown in Table 2. For the land price changes and total population changes in a given year indicated in the second row, a positive and statistically significant parameter was estimated only when the population was in directional decline. Columns 3, 4, and 5 show estimated results using 5- and 7-year moving averages. In estimates using the 7-year moving averages in column 6, the significance level is cleared at 1%. Thus, we adopted these parameters to project long term land price changes as in Equation (5).

**Table 1.** Summary of statistics.

| Variable Name | Mean | Std. Dev | Minimum | Maximum | Observations |
|---|---|---|---|---|---|
| Yearly growth rate of population | −0.00631 | 0.01168 | −0.07714 | 0.05979 | 5185 |
| Growth rate of population, 5-year moving average | −0.00627 | 0.01129 | −0.06744 | 0.05371 | 4281 |
| Growth rate of population, 6-year moving average | −0.00630 | 0.01125 | −0.06097 | 0.04953 | 4055 |
| Growth rate of population, 7-year moving average | −0.00633 | 0.01122 | −0.05647 | 0.04361 | 3829 |
| Yearly growth rate of land prices | −0.02478 | 0.04933 | −0.46745 | 1.51724 | 5185 |
| Growth rate of land prices, 5-year moving average | −0.02385 | 0.03075 | −0.15346 | 0.30345 | 4281 |
| Growth rate of land prices, 6-year moving average | −0.02325 | 0.02817 | −0.14261 | 0.25287 | 4055 |
| Growth rate of land prices, 7-year moving average | −0.02303 | 0.02558 | −0.13610 | 0.21675 | 3829 |
| Logarithm of land price/m$^2$ in 2000 | 11.4949 | 0.9993 | 8.0064 | 13.1149 | 5185 |
| Dummy variable of increasing population | 0.26500 | 0.44137 | 0 | 1 | 5185 |
| Dummy variable of decreasing population | 0.73500 | 0.44137 | 0 | 1 | 5185 |
| Dummy variable of big region $i$ | 0.223722 | 0.41678 | 0 | 1 | 5185 |
| Dummy variable of small region $i$ | 0.776278 | 0.41678 | 0 | 1 | 5185 |

**Table 2.** Estimation results.

| | Growth Rate of Land Price | | | |
|---|---|---|---|---|
| | Real | 5-Year Moving Average | 6-Year Moving Average | 7-Year Moving Average |
| Increasing Growth rate of Population | −0.180400 (1)<br>0.149980 | 0.18016<br>0.11017 | 0.265600 ** (2)<br>0.104745 | 0.35015 ***<br>0.09993 |
| Decreasing Growth rate of Population | −0.012565<br>0.073677 | 0.12159 **<br>0.05128 | 0.160380 ***<br>0.047910 | 0.19286 ***<br>0.04478 |
| Log of land price in 2000 (City > 500,000) | −0.001054<br>0.000125 | −0.00100 ***<br>0.00010 | −0.000900 ***<br>0.000100 | −0.00087 ***<br>0.00007 |
| Log of land price in 2000 (City < 500,000) | −0.002500<br>0.000090 | −0.00240 ***<br>0.00006 | −0.002310 ***<br>0.000060 | −0.00225 ***<br>0.00005 |
| Observations<br>Adjusted R-square | 5185<br>0.2235 | 4281<br>0.4145 | 4055<br>0.4502 | 3829<br>0.4889 |

Note: (1) Upper rows are parameters, lower rows are standard deviations. (2) *** and ** indicate that the null hypothesis is rejected at the 1 and 5% significance levels.

### 3.4. HAV Projection Method

The amount of housing asset was calculated by multiplying the housing land price per square meter of residential area, which was assessed for property tax by the concerned municipality. If the number of municipalities is kept $i$ and year $j$, then the amount of housing asset value in the Kansai metropolitan area can be expressed as in Equation (6). Nominal $V_{ij}$ is the price of land for residence, multiplied by the area of residential land. Nominal $V_{ij}$ does not consider price fluctuations; hence, it should be adjusted by the GDP deflator to calculate the amount of housing asset converted to real price in 2021. The

building value was not considered in this estimate because the building value of Japanese houses is almost worthless after nearly 20–30 years due to statutory years of durability.

$$V_j = \sum_{i=1}^{226} \frac{p_{i,j} * s_{i,j}}{g_j} \tag{6}$$

*V*: Housing asset value (JPY);
*P*: Standard land price for residential use (JPY/m$^2$, 1984–2021: real; 2022–2045: projected by Equation [5]);
*S*: Residential area on a property tax base by each municipality (m$^2$, 1984–2021: real; 2022–2045: assumed same as that in 2022 because there is almost no new development site in the Kansai metropolitan area);
*g*: GDP deflator as of 2021;
*i*: Municipalities (1–226);
*j*: Year (1984–2021: real; 2022–2045: projected).

## 4. Results

### 4.1. Projected Results

Figure 3 depicts the projected results: although real data are available until 2022, we calculate the difference between recorded values and estimated values beginning in 2018 in order to more accurately compare the evolution of the Kansai Metropolitan Area with Tokyo. The predicted total loss in the value of land used for housing in the region, by 2045, would reach a little more than 41.02 trillion JPY (around 298.3 billion USD) according to this simulation. It represents less than one half of the aggregate figures found for Tokyo with the same calculation (−94 trillion JPY, or around 667 trillion USD). The annual rate of decrease is still slightly superior, at 1.7% (against 1.3%). With regards to the history of the Tokyo–Osaka balance of power, we see that the residential markets of the Kansai region are already several orders of magnitude behind those of the capital region, indicating that regional gaps have broadened.

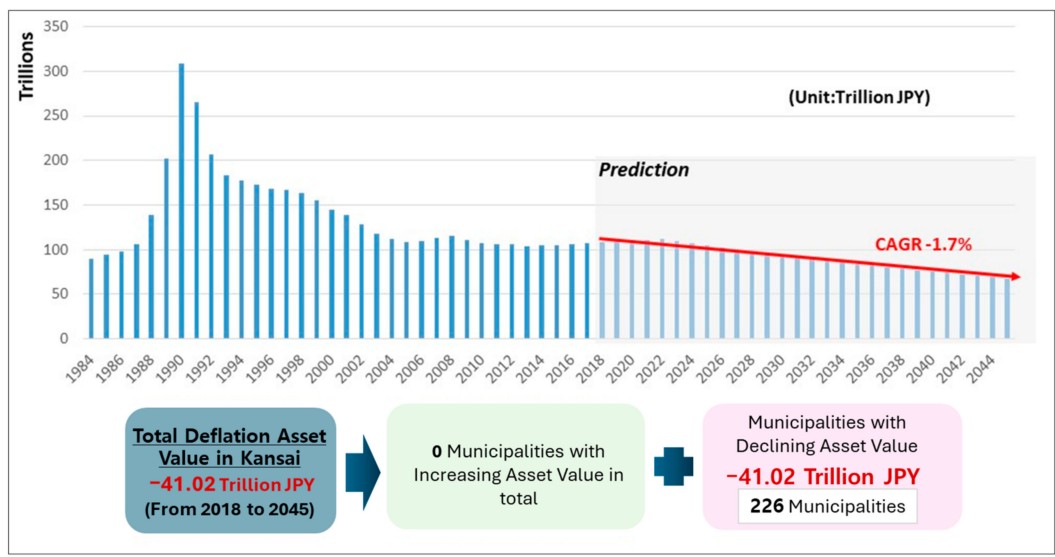

**Figure 3.** Evolution of total housing asset values in Kansai Metropolitan Area.

A mapping of total housing asset values on a municipal scale in 2018 and 2045 reveals an unequal distribution of wealth in favor of the region's main poles (Kyoto, Osaka, Kobe, Nara, Wakayama, and Himeji) and their surrounding suburbs, and this hierarchy is lasting (Figures 4 and 5).

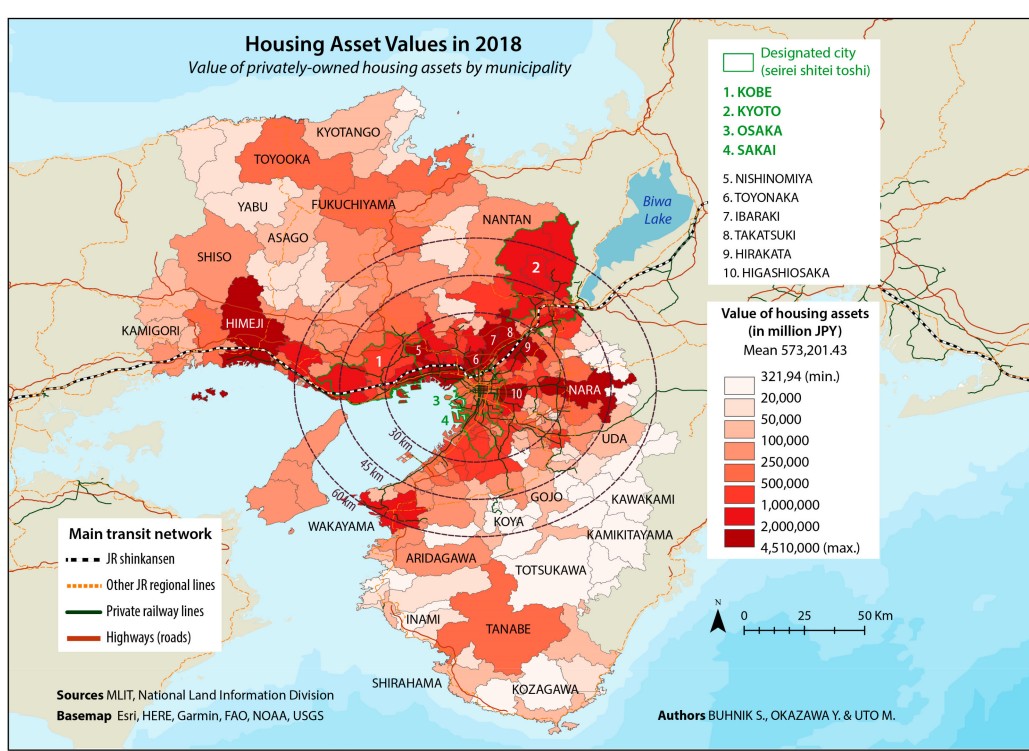

**Figure 4.** Total housing asset values by municipality (2018). Note: Range rings are centered on the CBD district of Honmachi, between Umeda and Namba, in Osaka City.

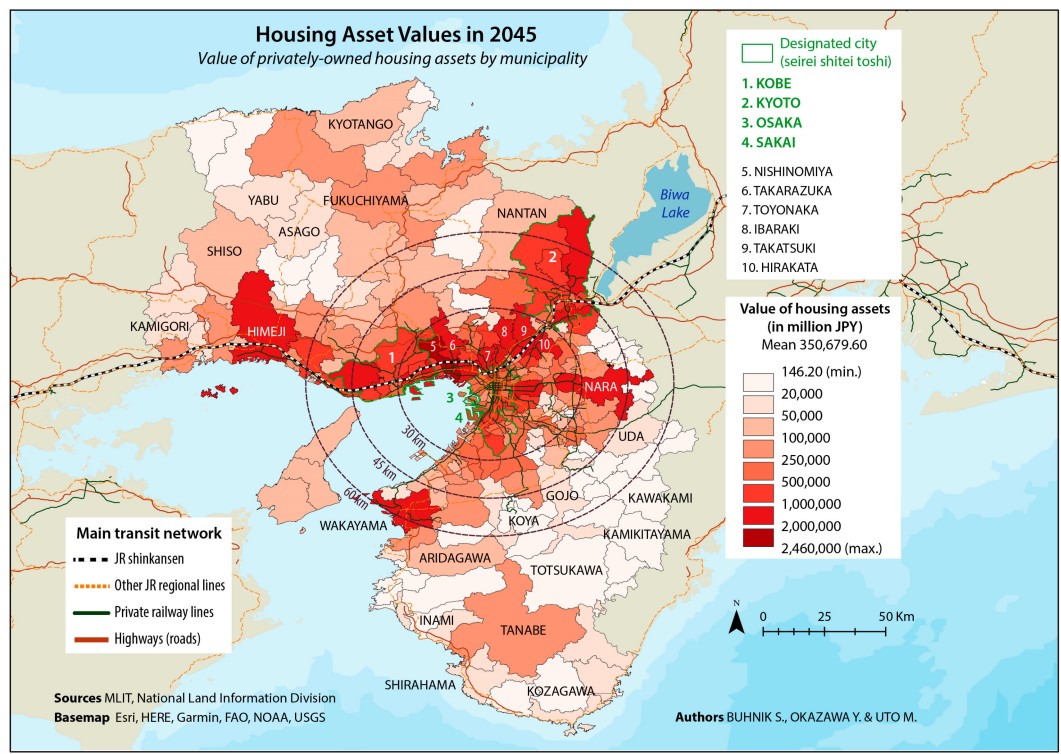

**Figure 5.** Estimated total housing asset values by municipality (2045).

The same can be said about municipal hierarchies in the Tokyo region; but what distinguishes Kansai is that (1) municipalities with the highest HAV are not all located within a 45 or even a 60 km radius from major employment cores (i.e., Osaka, Kyoto, Kobe);

(2) the HAV levels of Kansai's main poles do not seem considerably superior to several suburbs and even more distant municipalities. However, we must note that the HAV of many rural municipalities is augmented by their sheer geographical size, and that all cities over 500,000 inhabitants are divided into wards that are comparatively tiny. If we add up the value of all wards belonging to Osaka city in particular, the sum slightly exceeds 14 trillion JPY in 2018–2022, while some neighboring municipalities like Nishinomiya or Takatsuki are in the 2–5 trillion range (Figures 4 and 5).

Ref. [62] underscore how, after the stagnation of the 1990s, property values increased again in Tokyo city, owning to a renewed demand for housing and an accumulation of public/private investments backed by the 2002 Urban Renaissance Law. Such gentrification based upon condominium construction, albeit not negligible, has been more moderate in Osaka and primarily fixed itself in the city's northern wards, around the business hubs of Umeda and Shin-Osaka shinkansen station; recently, however, repeated attempts to rebrand Osaka's image through retail gentrification [63] have accompanied a transformation of the city's landscape around Nishi, Namba, and Tennoji. Residential redevelopments stirred by tourism and rehabilitations after the 1995 earthquake generated population gains for the cities of Kyoto, Kobe, and Sakai.

These background elements help us interpret the inter-municipal distribution of estimated HAVs by 2045, expressed as a ratio of their value in 2018 (Figure 6). According to our calculations, land for residential use in the region's urban cores, which remain the most demographically dynamic, could retain at least 70–80% of the value they have as of 2018. By contrast, surrounding municipalities would barely keep 50–60% of it (on the Osaka Bay and the Sea of Japan, like Kyotango) or even 40–50%, especially in the south-east of Osaka, the landlocked areas in Nara and Wakayama, and the western margins of Hyogo Prefecture.

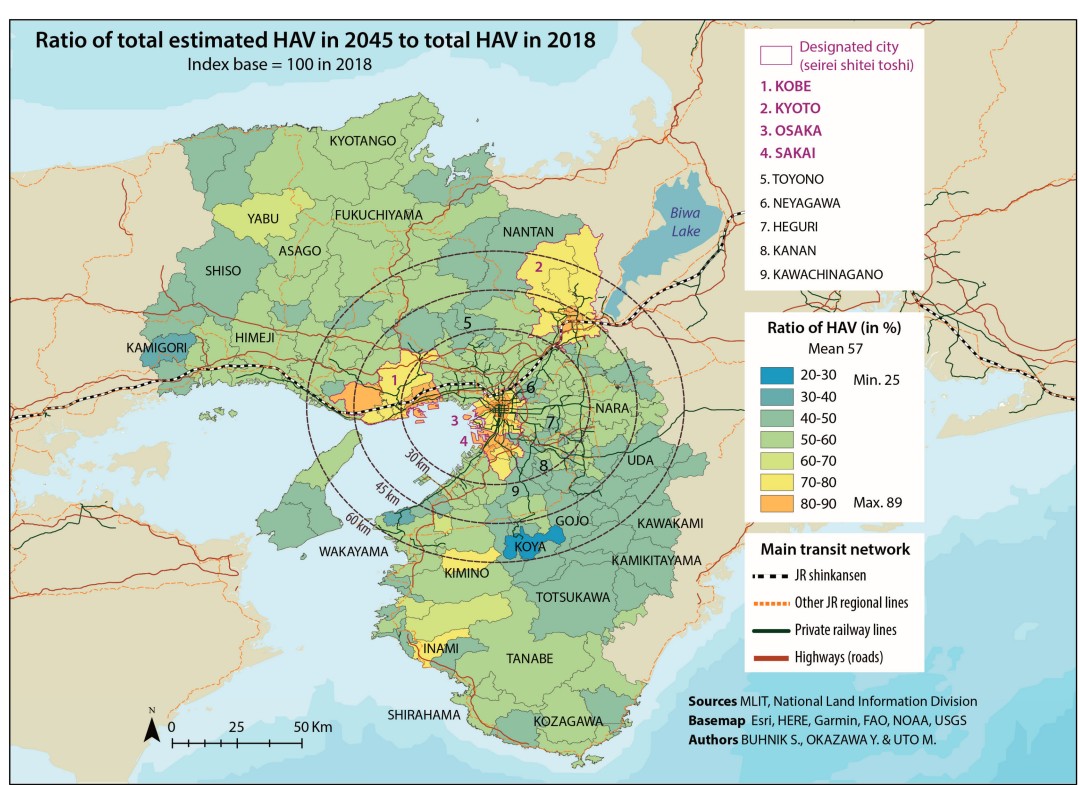

**Figure 6.** Ratio of total housing asset value (2045).

*4.2. Differences in Housing Asset Value Decline by Commuting Time and by Household*

Considering the results we mapped, we can provide some hints to the two questions raised at the beginning of Section 3. We first asked whether depopulation and decrease

in HAVs are distributed in similar patterns or not in Tier 2 (Kansai) and Tier 1 (Tokyo) metropolises; second, we wondered whether the effects of HAV deflation would be more severe in Tier 2 rather than Tier 1 (Tokyo) metropolises.

According to [55], in the case of Tier 1 (Tokyo Metropolitan Area), no municipality located beyond a 45km radius of Tokyo's central business districts should expect to retain more than 40–50% of their current housing asset values by 2045, so that spatial inequalities in the accumulation and de-accumulation of housing wealth would worsen. The overall picture in Kansai is more complex, since we can discern municipalities staying above the threshold of 60% beyond a 60 km radius. This is confirmed by our calculation of the average HAV deflation ratio by municipality, depending on the average commuting time by car or train necessary to reach core employment areas (Table 3). While the rate of loss regularly increases with the distance in the case of Tokyo, Kansai's biggest "losers" are municipalities situated within a 60–120 min range.

**Table 3.** Housing asset value changes by commuting time.

| Commuting Time by Car (Minutes) | Value of Housing Asset/Household (JPY) | | CAGR | Rate of Change |
|---|---|---|---|---|
| | 2018 | 2045 | | |
| 0–30 | 11,250,303 | 8,636,873 | 0.1% | −23.2% |
| 31–60 | 13,953,821 | 10,207,515 | −0.6% | −26.9% |
| 61–90 | 12,738,964 | 8,411,902 | −0.9% | −34.0% |
| 91–120 | 10,400,655 | 7,275,839 | −1.3% | −30.0% |
| 121–150 | 7,967,336 | 6,123,416 | −1.6% | −23.1% |
| 151– | 7,853,192 | 6,072,845 | −1.6% | −22.7% |
| Total | 13,043,024 | 9,351,649 | −0.7% | −28.3% |
| Commuting Time by Train (Minutes) | Value of Housing Asset/Household (JPY) | | CAGR | Rate of Change |
| | 2018 | 2045 | | |
| 0–30 | 14,979,154 | 11,528,334 | 0.1% | −23.0% |
| 31–60 | 13,706,874 | 9,905,336 | −0.6% | −27.7% |
| 61–90 | 13,049,082 | 8,966,526 | −0.9% | −31.3% |
| 91–120 | 11,996,452 | 8,100,518 | −1.1% | −32.5% |
| 121–150 | 9,002,952 | 6,761,183 | −1.5% | −24.9% |
| 151– | 7,830,355 | 5,906,598 | −1.5% | −24.6% |
| Total | 13,043,024 | 9,351,649 | −0.7% | −28.3% |

How can we explain that some outlying municipalities, according to our model, would be "faring better" than municipalities holding currently more inhabitants and economic activity? A main argument is that already by 2018, the average land prices of these areas are at a low level: their earlier and stronger pace of depopulation spurred housing vacancy and put an extreme downward pressure on the value of still-occupied residential land. To say it bluntly, the weaker value of housing assets in these areas is such that further losses would not be quantitatively important, both in volume and percentage of the HAV. Compared to Tokyo, the Kansai metropolitan area lacks a specific center while Tokyo has a dominant center, special wards, and dependent bed towns. Meanwhile, Kansai displays an Osaka–Kyoto–Kobe multi-centric city structure that makes commuting by car from rural areas easier than resorting to railway transportation, so that the influence of proximity to railway hubs on land prices is less influential than in Tokyo.

Finally, the financial implications, both in collective and individual terms, are addressed with Figure 7. Because they presently hold some of the most valued residential markets of the region but face stronger rates of aging and depopulation than regional cores, the municipalities along the backbone of the Kansai metropolitan area (like Nishinomiya, Takarazuka, Takatsuki) may face the greatest HAV deflation, as measured by the sum of the difference in JPY/m$^2$ between 2018 and 2045. Such municipalities have grown as bedroom communities for mainly more than upper middle-income households; thus, the residential land price is now relatively high compared with other residential areas. If we consider the

difference per household, an average loss of 7.5 to 13 million JPY (around 53,000 to 91,000 USD) per household could be expected.

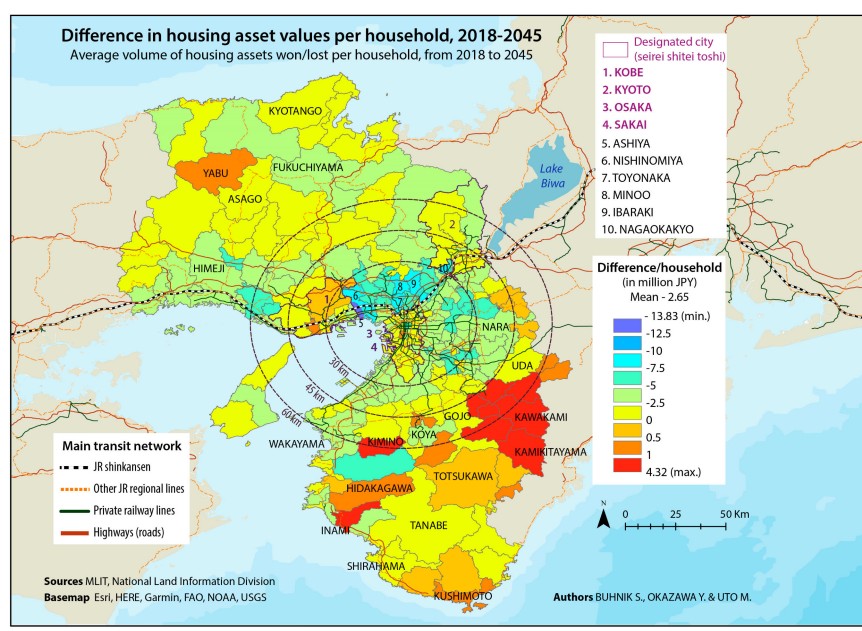

**Figure 7.** Differences in housing asset value per household.

*4.3. Contrasting Tier 1 vs. Tier 2 Metropolises*

Looking at the HAV estimation results for the Kansai and Tokyo metropolitan areas (Table 4), the population decline rate in the Kansai metropolitan area is −18.3%, which is much higher than the −6.6% in the Tokyo metropolitan area. In addition, the HAV decline rate is −37.7%, which is higher than that of the Tokyo metropolitan area. We found that the rate of HAV decline in Tier 2 Kansai is higher than that in Tier 1 Tokyo. On the other hand, the rate of decrease in the HAV by commuting time tends to increase in Kansai in proportion to commuting time. The maximum decrease is over −30% within 60–120 min, but the decrease rate after that is as low as −25%. In addition, the HAV decrease rate is −23% even within 30 min from the Kansai CBD, and the HAV decrease rate is also significant near the CBD. We found that in Tier 2 Kansai, compared to Tier 1 Tokyo, there is a greater HAV decrease nearer to its CBD, and the decrease rate of HAVs is more uniform overall in Tier 2 Kansai. A possible reason for this is that as the size of the metropolitan areas is smaller, the trend of decline also spreads to the vicinity of the CBD.

**Table 4.** Comparison of housing asset value changes in Kansai and Tokyo.

| | Tier 1<br>Tokyo Metropolitan Area | Tier 2<br>Kansai Metropolitan Area |
|---|---|---|
| **Population 2018 (people)** | 36,469,816 | 19,201,706 |
| **Population 2045 (people)** | 34,058,554 | 15,690,697 |
| **Rate of population change** | −6.6% | −18.3% |
| **Housing asset value 2018 (Trillion JPY)** | 322.5 | 108.8 |
| **Housing asset value 2045 (Trillion JPY)** | 228.5 | 67.8 |
| **Rate of HAV change** | −29.2% | −37.7% |
| **Rate of HAV change by train commuting time (minutes)** | | |
| **0–30** | −9.9% | −23.0% |
| **31–60** | −29.8% | −27.7% |
| **61–90** | −48.2% | −31.3% |
| **91–120** | −54.7% | −32.5% |
| **121–150** | −61.2% | −24.9% |

We considered differences in the spatial decreasing trends of the HAV. A trend commonly observed in both metropolitan areas is that municipalities developed as bedroom communities during the post-war population growth period faced a notable HAV decrease. In the Kansai metropolitan area, there is a concentration of bedroom communities in the northern part of Osaka, and such municipalities were exceeding 7.5 to 13 million JPY (around 53,000 to 91,000 USD, see dark blue and light blue areas in Figure 8). There are many bedroom communities in the western part of the Tokyo metropolitan area, and a similar decline in HAVs is observed in the Kansai metropolitan area. However, there is a difference in location trends: the Kansai metropolitan area has some bedroom communities within a 30 km radius, indicating a significant HAV decrease even in municipalities with short commuting times. On the other hand, in the Tokyo metropolitan area, where many bedroom communities are located outside the 30 km radius, the decrease in the HAV is more notable in municipalities with longer commuting times. Many municipalities in the Tokyo metropolitan area undergo larger HAV declines than those in the Kansai metropolitan area because of its more extensive bedroom communities. Since HAV decline in bedroom communities is large, it has a negative impact on the retirement life of elderly households, as pointed out by Uto et al. (2023) [55]. Considering this point, we believe that the impact of HAV decline is more likely to be a social problem in a Tier 1 metropolitan area like Tokyo, so that the effects of urban shrinkage will be more severe than in Tier 2 Kansai.

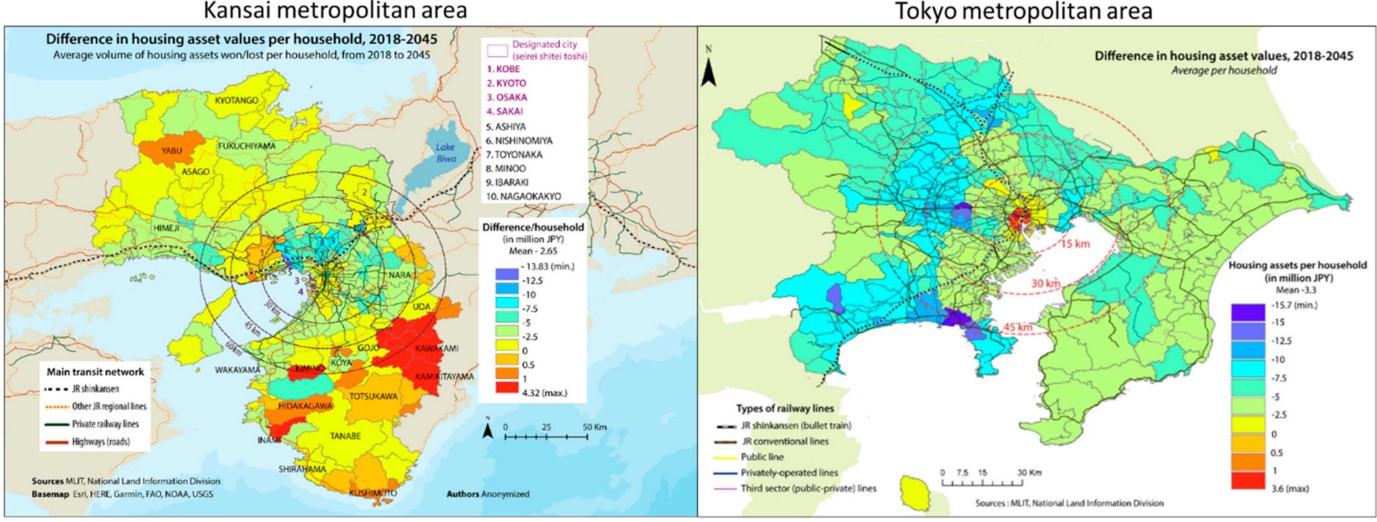

**Figure 8.** Comparison of housing asset value per household changes in Kansai and Tokyo.

Next, looking outside the 45 km radius, there is a trend of HAV decrease in the Tokyo metropolitan area, whereas HAVs are stabilized in the Kansai metropolitan area, and there is even a slight increase in some municipalities. We assumed that these differences in the size of the metropolitan areas are due to the fact that fewer households in the Kansai metropolitan area commute to the CBD from municipalities that are distant by 120 min or more (≈45 km outside the metropolitan area), and such areas are not affected by the population decline. In addition, municipalities outside the 45 km radius of the Kansai metropolitan area are either tourist destinations or residential areas where land prices are already quite low, so even if the population declines, there is little room for HAV decline.

As we have discussed, the Kansai metropolitan area has a higher rate of population decline and HAV decrease than the Tokyo metropolitan area, and municipalities closer to the CBD also tend to experience declines, with Tier 2 Kansai being more susceptible to population decline than Tier 1 Tokyo. On the other hand, the Tokyo metropolitan area has a larger number of bedroom communities, so the municipalities where decline in the HAV per household exceeds 7.5–13.0 million JPY (around 53,000 to 91,000 USD) are spatially distributed over a much wider area, while the Kansai metropolitan area only

has a concentration in northern Osaka. This also suggests that the problem of declining HAVs in the Tokyo metropolitan area may be more serious than in the Osaka metropolitan area. One of the issues we focused on was the correlation between income multipliers for the HAVs per household. In general, the larger the population of a metropolitan area, the higher the land price, but households in metropolitan areas also have higher incomes, which to some extent offsets the higher land prices. But if land prices are higher than income, the income multiplier is higher. If the income multiplier is higher, the ratio of real estate to household asset should be higher. Hence, we hypothesized that the impact of HAV decline on households in the Tokyo metropolitan area would be larger than in the Kansai metropolitan area. To examine this, we calculated the income multiplier of HAV per household in 2018. Since there are no statistics on household incomes for all municipalities, we only looked at major cities, but a comparison of the latter's income multipliers still shows that the Tokyo metropolitan area had higher income multipliers than the Kansai metropolitan area (Table 5).

**Table 5.** Comparison of housing asset deflation impact by major cities.

| Tier 1: Tokyo Metropolitan Area | | | | |
|---|---|---|---|---|
| **Major Cities** | Tokyo | Saitama | Chiba | Yokohama | Average |
| **Rate of depopulation** | 3.3% | −0.5% | −6.5% | −7.8% | −2.9% |
| **Rate of HAV deflation** | −20.6% | −18.0% | −17.4% | −19.1% | −18.8% |
| **Income multiplier** | 3.82 | 2.99 | 1.86 | 2.99 | 2.92 |
| **Tier 2: Kansai metropolitan area** | | | | |
| **Major Cities** | Osaka | Sakai | Kyoto | Kobe | Average |
| **Rate of depopulation** | −11.6% | −14.9% | −11.7% | −15.3% | −13.4% |
| **Rate of HAV deflation** | −18.2% | −8.7% | −10.5% | −4.7% | −10.5% |
| **Income multiplier** | 1.98 | 2.52 | 3.05 | 2.53 | 2.52 |

Note: Tokyo is 23 Special Districts; others are whole cities.

When comparing the average income in each area, Tokyo city holds the highest at 6575 thousand JPY [64]. The income multiplier was 3.82 in Tokyo city, while in Osaka, Kyoto, and Kobe city, it was 1.98, 3.05, and 2.53, respectively. This indicates the leverage to purchase a house by household. The Kansai metropolitan area appears to have a lighter burden when it comes to buying homes in comparison to the Tokyo metropolitan area. In Tokyo city, the HAV will decline −18.8% by 2045 against −10.5% in Kansai. The impact of HAV deflation would thus be a more severe problem, especially for Tokyo city, which presently holds higher housing values.

To examine the relationship between income multipliers and HAV declines, the results in Table 5 are plotted on the scatter plot of Figure 9. The correlation coefficient between HAV decline per household and income multiplier is −0.71, which shows a relatively strong negative correlation. This suggests that cities with higher income multipliers tend to have higher HAV declines per household. Cities with housing prices that are relatively high for the city's income level will experience greater housing asset deflation. This is consistent with the finding that bedroom communities experience greater housing asset deflation. If it were not for commuters to the CBD, bedroom communities would have had lower housing prices. As the city expanded, housing prices skyrocketed despite overly long commuting time. But as the population continuously declines, housing closer to the CBDs becomes more affordable, so that the relevance of bedroom communities comes to an end.

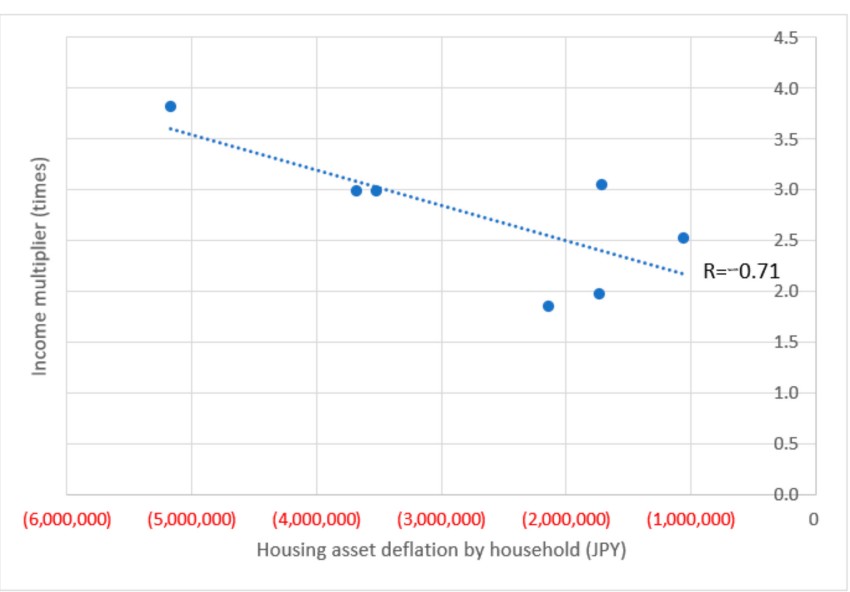

**Figure 9.** Relationship between income multiplier and housing asset deflation.

## 5. Discussion

The focus of this study was to examine the deflation of HAVs in the Kansai metropolitan area. In addition, by comparing the Kansai metropolitan area with the Tokyo metropolitan area, differences between metropolitan areas of different sizes (Tier 1 and Tier 2 classes) were examined.

These findings were discussed in a manner that eliminates as much as possible the elements unique to the Tokyo metropolitan area and the Kansai metropolitan area, because we thought that the value of this study would diminish if its results were not interpreted from a global perspective as well. This article addresses the socioeconomic challenges that population decline, and hollowing-out processes entail for metropolises, by assessing the evolution of housing asset values for each municipality belonging to the Kansai metropolitan area. Our estimates, which run until 2045, re-apply to this region a model recently elaborated by [55] to calculate the impacts of HAV deflation on the housing wealth accumulated so far by municipalities and residents of the Tokyo metropolitan area. We found that the HAVs in Kansai declined faster than Tokyo's (1.7% annually against 1.3%). But by 2045, it would amount to less than one half of the latter's expected losses (approximately 41 trillion JPY against 94 trillion). Hence, we can assert that disparities in the concentration of wealth associated with homeownership would widen between Tier 1 and Tier 2 regions. Furthermore, we confirm that suburban municipalities specialized in "dormitory" functions emerge as local losers from the current effects of demographic decline.

We pointed out that cities with higher HAV income multipliers tend to have higher HAV deflation. This indicates that cities that developed as bedroom communities during urban expansion periods benefited from higher HAV growth despite their greater distance from regional CBDs. This indicates that the HAVs of the more expensive bedroom communities may fall the most during periods of population decline. The highest HAVs for bedroom communities are in the most populous metropolitan areas, implying that bedroom communities in Tier 1 metropolitan areas will experience the largest HAV declines. On the other hand, Tier 2 metropolitan areas did not have such high HAV income multipliers, so that their HAV declines turn out to be lower than those of Tier 1 bedroom communities. Thus, we clarified that the HAV of both Tier 1 and Tier 2 bedroom communities would decline, albeit to a different degree.

This suggests that densely populated Tier 1 metropolitan areas such as Tokyo may face serious social problems and be unable to maintain their housing values in the future,

while Tier 2 metropolitan areas such as Kansai, albeit with lower HAVs, may experience greater stability in an aging society.

## 6. Conclusions

The relationship between HAVs and average income showed that households in Tokyo needed to earn 3.82 times said income to purchase a home, compared to Osaka, where the income requirement is 1.98 times. This higher leverage in Tokyo causes serious problems for elderly households due to HAV deflation, but lower leverage in Kansai may help stabilize the lives of elderly households. Considering our study, we can state that the Tier 2 metropolitan area was exposed to the effects of HAV deflation at an earlier and faster pace, but the Tier 1 metropolitan area encounters more serious social problems. Hence, our conclusion is that these are problems of differing nature rather than just assessing which one is more severely affected by HAV deflation.

The contribution of this study is to highlight the impacts of HAV deflation on residents living in metropolitan areas of various sizes, and to show that it always raises serious challenges, albeit different in nature. This point led us to imagine that even smaller metropolitan areas cope with different problems. We believe that these issues are worthy of continuing research. In addition, this study pointed out that differences in the size of metropolitan areas alters the problem of urban shrinkage. The diversity in metropolitan areas' sizes over the world indicates that the issue of urban shrinkage could not be discussed within a single, all-encompassing category of metropolitan areas. From a global perspective, the findings suggest that discussions need to be tailored to each of these various city sizes. In this sense, we believe that this study provides valuable insights.

In terms of scientific prospects, one key issue is to achieve a more accurate estimation: to do so, it is necessary to derive a more appropriate land price function and to try time series forecasting approaches such as ARMA, VAR, or STAR models. Furthermore, it is crucial to analyze the effects of HAV deflation in regions beyond Kansai and Tokyo, encompassing other categories of metropolitan areas, so as to enhance our comprehension of the variegated geographies of housing asset value changes in contexts of long-term depopulation.

**Author Contributions:** M.U.: Conceptualization, Methodology, Formal analysis, Revised and edited draft, Funding acquisition. S.B.: Data collecting, Formal analysis, Original draft, Revised and proofread draft. Y.O.: Data collecting, Formal analysis. All authors have read and agreed to the published version of the manuscript.

**Funding:** This study was supported by the Japan Society for the Promotion of Science (JSPS) KAKENHI (Grant Number 22K04500).

**Data Availability Statement:** Data are contained within the article.

**Conflicts of Interest:** Author Yuki Okazawa was employed by the company Mitsubishi Research Institute Ltd., Japan. The remaining authors declare that the research was conducted in the absence of any commercial or financial relationships that could be construed as a potential conflict of interest.

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
