# Peer review of "How Are Tier 2 Metropolises Affected by Housing Asset Value Deflation in the Depopulation Era? A Comparison between the Tokyo and Kansai Metropolitan Areas"

_land, doi:10.3390/land13040418_

Round 1
Reviewer 1 Report
Comments and Suggestions for Authors
This is a potentially interesting paper on a topic highly relevant for scholars of land and housing economics. The central question is how land values are declining (and may continue to do so in future) in small metropolitan areas that are facing population decline.
I understand that this paper borrows methods from the previous Cities paper, which I’ve now also looked at. However, I see major concerns with the regression model specification. I’m not sure why the reviewers for the Cities paper did not mention these issues:
· Dummy variables are usually specified at 1 for one subset of a binary group and 0 otherwise. Yet, you specify separate dummies for each part of the binary group. This should lead to perfect multicollinearity and an inability to solve the model. It also goes directly again the convention for the specification of dummies in regressions. Please explain.
· In this paper you discuss housing value change, but it seems you measure land value change. This is very confusing. It’s also not clear that you are using assessment data until late in the paper. It’s not clear as well how and why you aggregate data across the municipalities. What is the dependent variable of the regressions? Average assessment value per unit area? For housing or land or both? Do you have the individual observations with spatial geocodes? If so, why aggregate? In order to also look at population change as an explanatory variable?
· Even if you aggregate, you need to control for temporal autocorrelation, because you have a lag term. In fact you should run a STAR (space-time autoregression) model, as it’s well know that space-time autocorrelation is present. Again strange that reviewers did not suggest this.
· The second part of the paper goes into a lot of analysis and discussion of how household income and proximity to employment impact land value changes. Both should be included as independent variables in the regressions. Interest rate should also be included, if you are doing a panel regression.
· You predict until 2045, which to me is not valid. I can see predicting 5-10 years in the future, with caveats, but there is little reason to believe that population projections that far in the future are likely to be valid. You also don’t do any in-sample or out-of-sample validation or include a confidence interval around your projection (again, that was not asked in the Cities manuscript, which surprises me.). That confidence interval should technically also consider the variance of the population projections. I suggest using the open-source Map Comparison Kit to explore, in sample, where your model does well and where it does not.
The paper is currently also very long. I think by including employment access (maybe one variable for train transit and another for car transit, if they are not too closely correlated; otherwise include the minimum transit time or average of it), you could shorten the paper and discussion a lot.
Again, this is an important topic, and the paper could be a strong contribution, but the quality of the research design and the quality of the writing really need improvement.
Comments on the Quality of English LanguageThe English and overall clarity of language in the paper is extremely uneven, and so bad in places it was difficult to discern the authors’ meaning, especially with respect to tense (past, present, and future). The title and abstract are particularly bad. I suggest that you work with an English editor directly to improve the manuscript before resubmitting it.
Reviewer 2 Report
Comments and Suggestions for Authors
This is a remarkably complete and compelling paper. Its main problem is that its interesting insights are obscured by the poor English.
Comments on the Quality of English LanguageExtensive English editing is required.
Reviewer 3 Report
Comments and Suggestions for Authors
Analyzing and comparing the changes of Housing Asset Values for two different Tier Metropolitan areas in Japan are interesting and marginally contributes to the literature. The methodology is that of the same authors' work in 2023 for Tokyo metropolitan area. The paper's objective is worthwhile to investigate and the empirical results, in comparing, are interesting and provide further insights that real estate markets are all local.
I think the paper is ready for the publication with very minor editorial revisions. For example, the title could be more precise: "How are Tier 2 metropolises affected by housing asset value deflation under depopulation era? Comparison between the Tokyo and Kansai metropolitan areas"
Also, in the abstract, one could be much more precise and clear. For example, "Our study analyzes the differences and similarities between Tier 1 (Tokyo) and Tier 2 (Kansai) metropolitan areas due to shrinking cities problems. Both of metropolitan areas will dramatically decrease the Housing Asset Value (HAV) of residential communities."......
I would also include some major empirical findings in the Introduction.
Comments on the Quality of English LanguageI think the paper is ready for the publication with very minor editorial revisions. For example, the title could be more precise: "How are Tier 2 metropolises affected by housing asset value deflation under depopulation era? Comparison between the Tokyo and Kansai metropolitan areas"
Also, in the abstract, one could be much more precise and clear. For example, "Our study analyzes the differences and similarities between Tier 1 (Tokyo) and Tier 2 (Kansai) metropolitan areas due to shrinking cities problems. Both of metropolitan areas will dramatically decrease the Housing Asset Value (HAV) of residential communities."......
Round 2
Reviewer 1 Report
Comments and Suggestions for Authors
You did not address my questions or suggestions for methodological changes:
Dummy variables are usually specified at 1 for one subset of a binary group and 0 otherwise. Yet, you specify separate dummies for each part of the binary group. This should lead to perfect multicollinearity and an inability to solve the model. It also goes directly again the convention for the specification of dummies in regressions. Please explain.
Even if you aggregate, you need to control for temporal autocorrelation, because you have a lag term. In fact you should run a STAR (space-time autoregression) model, as it’s well know that space-time autocorrelation is present. Again strange that reviewers did not suggest this.
· The second part of the paper goes into a lot of analysis and discussion of how household income and proximity to employment impact land value changes. Both should be included as independent variables in the regressions. Interest rate should also be included, if you are doing a panel regression.
